# Extension Networks and Dissemination of Horticultural Advancements: Development and Validation of a Professionalization Instrument

Kevan W. Lamm [1],*[ID], Alexa J. Lamm [1][ID], Kristin Davis [2], Catherine Sanders [1][ID], Alyssa Powell [3] and Jiyea Park [1][ID]

1 Department of Agricultural Leadership, Education, and Communication, University of Georgia, Athens, GA 30602, USA
2 International Food Policy Research Institute, Pretoria 0028, South Africa
3 RaceTrac, Atlanta, GA 30339, USA
* Correspondence: kl@uga.edu

**Abstract:** Horticulture is a vast scientific discipline ranging from ornamentals to consumable food, which is constantly evolving. One of primary goals of horticultural innovation is to improve consistency, and predictability, among products. Extension is one of the primary channels connecting innovations and technologies to growers. However, despite the importance of extension in the dissemination of horticultural advancements, there are no standards for the professionalization of extension networks. Therefore, there is a current gap in the ability to ensure consistency amongst extension providers through professionalization at the network level. The goal of the study was to develop and validate an extension professionalization scale to empirically measure the most critical factors associated with extension professionalization within extension networks. Methodologically, the study extends upon previous research which identified specific capacities associated with extension professionalization at the network level. Specifically, an exploratory factor analysis was undertaken to examine the latent factor structure of the previously identified items. The results of this study identified two primary factors associated with extension professionalization in networks: (a) awareness of the need for extension professionalization, and (b) the operational integration of extension professionalization activities at organizational levels. Although there is existing literature examining professionalization, there are no such instruments specifically developed within an extension context. The present study provides an original and novel tool to prepare more rigorously and consistently trained extension professionals to serve and support the horticulture industry.

**Keywords:** horticulture; international extension network; professionalization capacity

## 1. Introduction

### 1.1. Extension Networks Overview

Extension providers support farmers, including horticultural producers, by disseminating research-based techniques to enhance agricultural production and address stakeholder needs [1,2]. However, despite previous research which has examined extension network characteristics such as knowledge management [3], there has been little research examining the connection between knowledge source and perceived ability of extension practitioners to successfully convey such information to clientele. Ongoing professional development allows extension professionals to provide relevant information and services for their clientele [4,5]. However, limited funding, insufficient coverage of advisory services, and low literacy rates have led to inconsistent extension services for international clientele, which contributes to fragmented extension networks [2,6,7]. Thus, a fundamental challenge remains for extension: developing professional standards appropriate for the dynamic, complex environment of horticulture and agricultural more generally [2].

### 1.2. Horticultural Innovation Overview and Literature Review

In past ten years, multiple studies have documented innovations and technology in horticulture, in the United States, and internationally [8–13]. In the existing literature, researchers typically have addressed how new innovations and technology will benefit the horticulture industry. For example, within the context of Australian horticulture.

Significant innovations in horticulture and horticultural science have taken place through plant breeding, plant biotechnology, production system innovations, environmental management, improvements in media and fertilisers, irrigation design and protected cropping, plant health, integrated pest management, postharvest protocols and improved market access, to name a few [14] (p. 1131).

However, across the globe there is a range of horticultural production needs from the sophisticated state of the industry described above, to more fundamental challenges regarding basic food safety and quality control systems in other countries [15]. As horticultural production continues to evolve and become more sophisticated, there is a risk for growers in different contexts to be limited in their ability to capitalize on such advancements [5].

### 1.3. Professionalization Overview and Literature Review

The concept of professionalization encompasses the establishment, dissemination, and enforcement of the knowledge, standards, and qualifications which elevate a field to a profession [16]. Abbott [17] explained the core of a profession was "the special relation between client and professional, and the core of professionalization was the evolution of guarantees for this relationship" (p. 356). This conceptualization is pertinent for extension networks, based on the foundational structure of the client-professional (grower-extension agent) relationship [18].

Professionalization relates to individuals within an organization and the structure and management required for the acquisition and maintenance of power [19,20]. Professional power refers to an occupation's formal structure and power held by practitioners through social exchange with clients [19]. A professional represents both their local institution and professional group at the system level and must act with a shared set of knowledge, skills, and values which withstand scrutiny [21]. Professional autonomy, which assumes a professional possesses the relevant knowledge and skills for a practice, requires a foundational set of standards to clarify the values, priorities, and knowledge shared among professionals in the same field [21,22]. Professional autonomy for extension providers requires a balance between autonomy from the client—in which providers are not ultimately controlled by the clientele's notions of needs—and autonomy from a facilitating organization, or network [23]—where an professional is not constrained by the controls and demands of others [24]. Finding this balance is intended to support practitioners to make decisions without external pressures from non-professionals nor the employing organization [19,25]. In general, a professional should be empowered to do what is right for a client without undue control from the organizational structure, but with the requisite knowledge and authority to make such decisions with sound judgment.

Within the literature, it is unclear whether professionalization consists of specific predisposed characteristics of a profession, or whether professionalization is a method of image building developed for the purpose of acquiring and maintaining power. Forsyth and Danisiewicz [19] argue both concepts account for varying levels of autonomy within the professional domain. To explore these relationships, they developed a model of professionalization, which consists of three phases: (a) potential, (b) formulation, and (c) stabilization [19].

### 1.4. Extension Network Professionalization and Horticulture Integration

In parallel to the ongoing technology changes within the horticultural industry there is a simultaneous shift in the role of extension practitioners. For example, the role of public versus private service providers, both representing their services as extension has expanded [26]. Although not necessarily a negative trend, the shift does represent a funda-

mental tenant of professionalization, specifically, what entity is providing oversight of such identifications? Serving in the role of innovation intermediary has been well established as that of extension in the literature [27]. However, the proliferation of information availability through sources such as the internet, seed companies, and so forth [14] has resulted in an environment where there is limited ability to differentiate accurate versus inaccurate information, or information which may be motivated beyond grower success. Ultimately horticultural producers must be able to evaluate horticultural practices and determine if, and how, they choose to adopt them [28]. How can development of rigorous professionalization standards be achieved in a dynamic network with diverse actors, interests, and needs? The nature of agriculture, and horticulture in particular, is one of frequent change and innovation. The role of effective, and appropriately professional, extension personnel to support these endeavours is paramount [5].

### 1.4.1. Extension Network Professionalization Potential in Horticulture

Professionalization potential involves a client-serving occupation (e.g., extension) establishing professional status. To do so, both the predisposing characteristics of the occupation and the need for image building within the occupation must be considered [19]. It is also necessary to examine the nature of the services provided through three predisposing characteristics: essential (service is critical for clients), exclusive (practitioners have a monopoly on the task), and complex (service is not routine and requires specialized knowledge). Specialized knowledge and consideration of the image building activity are key, particularly within extension [17,29]. Previous research has found horticulture growers do perceive extension to be an essential source of information. For example, extension-based radio programs have been found to improve of adoption of horticultural practices such as water conservation and appropriate chemical usage [30]. The exclusive nature of extension knowledge is somewhat less clear given many emerging sources of information available [14]; however, the complex nature of horticultural information [31–33] also represents a challenge in distilling that which is pertinent from that which is not [14].

### 1.4.2. Extension Network Professionalization Formulation in Horticulture

Formulation involves public evaluation of the occupation's claim to professional status and the creation of a professional autonomy [19]. For professional autonomy to occur, the public must recognize the occupation provides essential, exclusive, and complex services. With a public-serving occupation, such as extension, this step is critical to enact the perception of professionalization with clientele [12,19]. For example, previous research has found attitudes toward horticulture-based extension programs were predicted by participant perceptions of extension personnel [31]. Furthermore, provision of private extension services in Kenya also involved a training program as a fundamental component of the process and ensure public evaluation of programs as necessarily rigorous [26].

Despite the importance of formulation, many previous studies have indicated a lack of formalized training among extension providers, has resulting in decreased horticultural technical competence. Specifically, "The differences between the expected and existing levels in technical competencies in horticultural crops suggest that the AEOs [Agriculture Extension Officers] need in-service training" [32] (p. 134). Similar observations were made amongst horticultural extension personnel as it related to pest management and surveillance [34]. The need for the formalization must therefore be recognized and established at the organizational, network, level [19].

### 1.4.3. Extension Network Professionalization Stabilization in Horticulture

Stabilization provides an additional opportunity for image building [19]. In this stage, occupations which have successfully exhibited their autonomy are perceived as true professions. These occupations may now use their power to legitimize their image in both the public and political sector and compete for limited resources from governing institutions [35]. As noted previously, the range of extension network professionalization

stabilization in horticulture presents a challenge when establishing general criteria. For example, in the United States the Extension Master Gardener program has existed for decades and has trained thousands of volunteers [36], the stabilization of the profession is therefore well established. These results are in contrast to horticultural extension worker research in Ethiopia which found, "relatively lower educational level of AEWs [agriculture extension workers] who do not have enough pesticide hazard related knowledge and cannot [ . . . ] practically advising [sic] on pesticide related hazards" [33] (p. 5). Professionalization must therefore become integrated into the functioning of the institution to ensure stabilization [19].

*1.5. Study Purpose*

Public perception of extension as a profession in horticulture requires capacity development and multifaceted strategies. To address the unique needs for extension network professionalization capacity development, Lamm et al. [2] recommended the development of a scale to evaluate current capacities of extension providers from a uniform perspective.

The purpose of this study was to develop and provide preliminary validation of an empirical instrument to measure the professionalization capacity of extension networks. Specifically, the goal of the research was to establish content validity, internal structure validity, response process validity, and consequential validity for a proposed the extension network professionalization scale.

## 2. Materials and Methods

*2.1. Study Background*

It is important to note the data used within this study were collected as part of a larger, global, capacity assessment project. Within the larger project several different extension network characteristics were collected and analyzed, professionalization was one such characteristic of interest. This disclosure is provided based on recommendations within the literature [37]. Additionally, the overall methods employed for this study are identical to the one described in detail in [3]. Based on recommendations in the literature [38], a summary of the methods is included below; however, for a more detailed description, readers are strongly encouraged to review the seminal work see [3].

The population for this study consisted of extension leadership and board members from extension networks across regional, sub-regional, and country divisions. A convenience sample was drawn from nine participating extension networks including regional networks representing Africa, the Caribbean, the Pacific Islands, Latin and South America, a sub-regional network representing West and Central Africa, and country level networks representing Kenya, Malawi, Nigeria, and Uganda.

*2.2. Data Collection*

The researchers collected data via a combination of online and paper-based questionnaires between June and December 2016. Serving as a pilot, the paper-based questionnaires were distributed to and collected from 43 participants from three networks, with five to sixteen responses per network. The paper-based questionnaires were used to establish response process validity, described in greater detail below. The remaining responses were collected via the Qualtrics online survey system using Dillman et al.'s [39] Tailored Design Method approach. The Tailored Design Method has been demonstrated to decrease measurement errors and increasing survey response rates [40–42]. In the current study, the data collection process included a pre-notice message sent to potential respondents prior to the beginning of data collection from an organizational leader.

Next, an email invitation to complete the survey was sent to all participants approximately two days later. A series of three reminder messages were sent every three to five days until the survey closed to all respondents. Of the 128 individuals invited to participate across both the pilot and primary study, there were a total of 122 completed surveys, resulting in a 95% response rate. The number of respondents was deemed acceptable to

complete the proposed analysis based on existing guidelines within the literature see [43]. It is important to note that due to incomplete responses, some individual items or indices may have lower response rates. Once compiled, the resulting data were analyzed using SPSS v26.

### 2.3. Instrument Item Development

To measure the levels of professionalization within extension networks a series of actions were undertaken to develop the scale items. First, an extensive literature review of existing scales and literature related to professionalization was conducted. Second, the results of the Lamm et al. [2] extension professionalization study served as the foundation for the scale. Based on the preliminary actions, a set of 12 researcher-developed professionalization items were developed. Scale item responses were rated on a four-point, Likert-type scale where possible responses ranged from 1 = *little to no capacity*, 2 = *some capacity, but very limited*, 3 = *good capacity, but could still be improved*, and 4 = *exceptional capacity, no need for improvement*. Additionally, respondents had the option to rate an item as *N/A-not applicable or no knowledge* if they had no knowledge of the item.

### 2.4. Instrument Validity

To establish validity for the scale, several methods were employed based on recommendations in the literature (see [44–46]). Prior to the development of the scale, a thorough and extensive review of the literature was conducted to ensure content validity. To establish response process validity, a panel of experts were invited to rigorously review the instrument. The panel members had expertise in scale development, extension, evaluation, and research methodology. Additionally, response process validity was established based on a pilot of the instrument where responses were collected using a paper-based version of the instrument administered in person. Following the completion of the instrument respondents were asked to provide feedback regarding the readability and interpretability of scale items and directions. Only minor adjustments such as wording updates and clarifications were made following the pilot process.

To establish internal structure validity, descriptive statistics, including skewness and kurtosis values, were calculated and reviewed to determine the characteristics of individual scale items. Additionally, a Cronbach's alpha coefficient was computed as a measure of internal consistency and reliability. To examine the nature of the factor structure as well as the observed data of the scales, an Exploratory Factor Analysis (EFA) was performed on the scale items. A Kaiser-Meyer-Olkin Measure of Sampling Adequacy (KMO) and Bartlett's test of sphericity were calculated to establish suitability of the scale items for EFA. In the EFA, latent variables were extracted using an orthogonal varimax rotation with a Kaiser normalization based on recommendations within the literature to improve clarity of observed results [43]. Eigenvalues greater than 1.0 and individual factor loadings with absolute values greater than 0.500 were retained [47].

Consequential validity was established through a follow-up survey distributed to 15 Global Forum for Rural Advisory Services members following the primary data collection and analysis. A total of 14 responses were obtained resulting in a 93% response rate. Consequential validity was established by asking respondents to indicate whether the results associated with the scale were useful and whether they intended to use the results to modify their networks (e.g., [45]).

## 3. Results

Prior to conducting the EFA on the scale items the KMO measure of sampling adequacy and Bartlett's Test of Sphericity were conducted. The observed KMO value was 0.838 and the Bartlett's test yielded significant results ($\chi^2$ = 516.70, $p$ < 0.001), both of which indicated suitability for further factor analysis. Based on orthogonal varimax rotation of the data, two factors were extracted and accounted for 62.086% of the cumulative variance. The results of the EFA are presented in Table 1. There were two items which were cross-loaded

between the two extracted factors. The EFA was repeated with the two items removed as a secondary analysis. The second EFA with the 10 remaining items, had an observed KMO value of 0.840 and the Bartlett's test yielded significant results ($\chi^2 = 365.12$, $p < 0.001$), indicating suitability for further analysis. The more parsimonious ten item scale again had two factors extracted and accounted for 62.486% of the cumulative variance. Based on the nature of the two extracted factors, two brief descriptions were proposed. The first extracted factor was identified as operational integration of professionalization in extension networks, or operational integration. The second extracted factor was identified as extension network awareness of professionalization, or awareness. The two extracted factors were independently analyzed using EFA procedures.

**Table 1.** Exploratory Factor Analysis of Aggregate Professionalization Scale.

| Scale Items | Factors | |
| --- | --- | --- |
| | 1 | 2 |
| Sufficient funding to support professionalization activities is present (PROF10) | 0.851 | |
| A monitoring and feedback loop where insights are used to inform future professionalization activities is present (PROF9) | 0.784 | |
| Network professionalization supports relevant to clientele (PROF11) | 0.766 | |
| The network has a clear set of messaging around RAS professionalization developed (PROF1) | 0.739 | |
| The network supports the identification of the resources needed to be successful within RAS (PROF8) | 0.726 | |
| Identifiable impacts associated with the network's professionalization efforts are present (PROF12) | 0.685 | |
| The network offers opportunities to enhance knowledge of educational practices (including educational methods and program development expertise) amongst clientele (PROF5) | 0.648 | |
| The network is aware of existing strengths and weaknesses within the RAS system (PROF6) | | 0.792 |
| Members of the network advocate for RAS professionalization (PROF3) | | 0.749 |
| The network offers an understanding of rural advisory services (PROF7) | | 0.650 |
| RAS professionalization activities align to the network goals (PROF2) * | 0.632 | 0.537 |
| Activities are directed towards building leadership capacity (including strategy development and managerial skills) amongst clientele (PROF4) * | 0.561 | 0.556 |

Note: Principal Component Factors. Blanks represent absolute loading values <0.500. Item identifiers in parentheses. RAS—Rural Advisory Service. *—Cross loaded item.

The first extracted factor, operational integration, consisted of seven items. Following the EFA, one latent variable was extracted and accounted for 60.054% of the total variance. The variable was associated with an eigenvalue of 4.204. The KMO value was 0.836, which indicated further factor analysis was warranted. Additionally, the Bartlett's test yielded significant results ($\chi^2 = 285.590$, $p < 0.001$), which justified further factor analysis.

The second factor, awareness, consisted of three items. Following the EFA, one latent variable was extracted, which accounted for 58.422% of the total variance and was associated with an eigenvalue of 1.753. The KMO value was 0.640 and the Bartlett's test yielded significant results ($\chi^2 = 41.280$, $p < 0.001$). Both criteria indicated that further factor analysis was warranted.

*3.1. Scale Reliability and Correlations Data Collection*

Following the EFA, descriptive statistics and measures of internal consistency for the two extracted factors were calculated. The results are presented in Table 2. Internal structure validity was analyzed based on indicators of normal response distributions including skewness and kurtosis. Each factor had a skewness values less than two and kurtosis values less than seven, indicating acceptable response distributions (see [48,49]). Additionally, Cronbach's alpha coefficients were calculated for both factors. Based on

accepted thresholds within the literature (see [50–53]) both factors had acceptable observed alpha values indicating acceptable internal consistency.

**Table 2.** Professionalization Scale: Descriptive Statistics and Scale Reliability.

| Factor | *N* | *M* | *SD* | Skewness | Kurtosis | Cronbach's α |
|---|---|---|---|---|---|---|
| Op Integration | 83 | 2.232 | 0.614 | 0.199 | −0.310 | 0.888 |
| Awareness | 107 | 2.997 | 0.509 | −0.487 | 0.334 | 0.641 |

Next, a Pearson correlation was calculated to examine the nature of the relationships between factors. An observed correlation of $r = 0.477$, statistically significant at the $p < 0.001$ level, was observed. The correlation between the operational integration and awareness factors was considered moderate using the Davis see [54] convention for interpretation of correlations, providing evidence against multicollinearity.

### 3.2. Consequential Validity

After analyses were completed, representatives from extension networks around the globe were provided a summary of results. Respondents were asked to provide responses as the value they would associate with the results provided by the professionalization scale. There were 91% of respondents who agreed or strongly agreed the results associated with the professionalization scale were useful. Additionally, 85% of respondents indicated they would try to use the professionalization information to modify their networks, 77% of respondents intended to use the professionalization information to modify their networks, and 75% of respondents expected to use the professionalization information to modify their network.

## 4. Discussion

This study sought to validate an empirical instrument measuring perception of professionalization capacity in extension networks, with a particular focus on horticulture. The strong relationship between horticultural innovation adoption and extension providers has been well established in the literature see [14]. Instrument validity was confirmed by assessing the instrument's content validity, response process validity, internal structure validity, and consequential validity. An exploratory factor analysis was performed on the proposed scale, and the findings were used to identify extracted factors. Additional exploratory factor analyses were performed on each resulting factor to further analyze resultant structures. The current study extends upon previous extension related research [3] and provides an additional perspective related to extension service provision. A recommendation for future research and practice would be to examine how different aspects of extension service provision interact and ultimately deliver more value-added services to clientele. For example, does effective knowledge management [3] serve as an entry condition for extension professionalization? Or does professionalization develop in tandem with effective knowledge management [19]?

Based on the existing literature within the professionalization domain, it was initially hypothesized three primary factors may emerge [19], specifically, (a) professionalization potential, (b) professionalization formulation, and (c) professionalization stabilization. However, the results of the exploratory factor analysis within the present study revealed the items in the aggregate scale only loaded on two factors: extension network awareness of professionalization, or awareness, and professionalization in extension networks, or operational integration. Despite the differences between the hypothesized and observed results from a factor perspective, upon further examination, the results of the present study tend to support and validate those of Forsyth and Danisiewicz [19].

### 4.1. Awareness of Extension Network Professionalization Potential in Horticulture

Professionalization potential refers to when a client-serving occupation, in this case extension, establishes professional status through image building and consideration of predisposing characteristics [19]. Each item in the awareness factor is therefore applicable to the potential concept. Specifically, extension networks which offer an understanding of extension services can communicate to clientele, the public, and other stakeholders about why extension is essential, exclusive, and complex. Communicating these unique characteristics of extension are crucial in the recognition of extension as a profession. Equally important to potential is establishing the image of a client-serving occupation as a profession. To create a public image of extension as a profession, networks must be aware of the strengths and weaknesses in their individual networks and the overall system of extension [17]. These results are consistent with previous research within the horticultural industry where extension practitioners needed to be recognized by clientele for their knowledge and expertise to improve innovation adoption [26,31,32].

### 4.2. Operational Integration of Extension Network Professionalization Formulation and Stabilization in Horticulture

Professionalization formulation refers to the phase where the public evaluates an occupation's claim as a profession and the creation of professional autonomy within an occupation [19]. In this stage, it is crucial for the public to recognize and acknowledge the occupation offers essential, exclusive, and complex services [19]. One way extension may be able to demonstrate the essential, exclusive, and complex nature of associated services is by developing a clear set of messaging which explains the motivation for the professionalization of extension. Through this messaging, networks are better enabled to communicate to clientele and stakeholders on how extension professionalization activities align with existing network goals and how support for professionalization of their network is relevant to clientele. Using examples of the benefit of extension efforts to support horticultural grower needs, such as mushroom cultivation [13] or technology adoption [9] may help to further demonstrate the relevance of extension to horticultural production.

Additionally, extension personnel can identify and communicate the impacts associated with the network's professionalization efforts. The value of extension as a profession can be tangibly demonstrated to the public, clientele, and stakeholders by directing activities towards enhancing knowledge of educational practices (e.g., educational methods and program development expertise) amongst clientele [55]. Furthermore, within the formulation phase, the occupation creates professional autonomy. An extension network may therefore develop autonomy by identifying resources necessary to be successful, which enable clientele and stakeholders to understand the inputs required for partnership [4]. In addition to continued clear and consistent messaging, the network may then be able to use services to demonstrate professional autonomy among the public.

Professionalization stabilization refers to the additional opportunity for image building and is the phase where occupations which exhibit their autonomy are perceived as professions [19]. Establishing monitoring and feedback loops where insights can inform future professionalization activities is a critical condition in this phase. Extension networks are thus encouraged to continue to maintain awareness of public perception and be amenable to receiving feedback. By maintaining this feedback loop, the network may better determine which services are desired by clientele and create clear and consistent communication which advocates for the professional autonomy of extension [14].

### 4.3. Limitations

While the results of this study are promising, there are several noteworthy limitations which must be addressed. First, the data were collected only in international extension settings located in the global South (e.g., Africa, the Caribbean, the Pacific Islands, and Latin America), which limits generalizability of the results. An associated recommendation is for future research to include larger and more diverse samples to improve scale robustness.

Additionally, a larger and more inclusive sample will provide insights to professionalization capacity of extension networks located in areas outside those surveyed within this study. A larger sample size would also provide additional statistical power to further analyze the structure of the proposed scale. Specifically, confirmatory factor analysis is recommended to further establish internal structure validity for the extension professionalization scale. Additionally, the proposed scale may be adapted to specifically focus on horticulture related content. As written, the scale is intended to establish a baseline assessment of extension network professionalization capacity across agriculture more generally. A more tailored version of the scale, focused on horticulture, may provide additional insights.

Another limitation associated with the scale is it only measures perceptions of professionalization capacity within extension networks, not empirical quantities of professionalization capacity. Although the use of perception-based scales is well established within the literature ranging from personality (e.g., [56]) to extension knowledge management capacity (e.g., [3]), the use and interpretation associated with scale results should be done with caution and from this perspective. An associated recommendation would be to use the results of the extension network professionalization scale as a starting point for preliminary needs assessments, or baseline data collection. The data may then be helpful to further investigation of individual extension network analyses related to professionalization.

An additional limitation is related to the observed Cronbach's alpha value associated with the awareness factor. Generally, Cronbach's alpha values over 0.70 are considered acceptable within social science research [50–52]. However, other scholars have proposed values of 0.64 or greater as adequate [53]. Based on this guidance the results of the present study were deemed acceptable, but certainly warrant additional investigation. As recommended previously, replicating the study with a larger sample to increase the associated statistical power associated with subsequent analysis may provide additional insights. Furthermore, adding additional items associated with the awareness, and by association professionalization potential [19], factor area may increase the overall internal consistency among items.

### 4.4. Contributions to Horticultural Practice

The volume of current horticultural advancements and technological innovations provides an important context for the criticality of innovation intermediaries [27]. Historically, extension providers have served in this role, diffusing such information to horticultural producers. Simultaneous to the volume of technological advancements is a fundamental change in the provision of extension services around the globe [5]. Without forward looking strategies to adapt and support the new paradigm the role of traditional extension services may be in jeopardy, with potential negative effects for horticultural producers [32,34]. From a systems-based perspective it is important to look not only at simplistic cause and effect interactions, but to instead look for circular effects which may only manifest as a result of a distal cause. In horticultural production plant growth is not only impacted by water and light the day before harvest but is a result of soil preparation before a seed is even planted. Similarly, extension networks are encouraged to consider using the proposed professionalization scale and to begin the process of raising awareness and integrating professionalization activities within their operations. Such efforts may help to improve consistency and rigor among extension personnel, providing benefits for clientele, including horticultural producers.

### 5. Conclusions

The present study established a robust quantitative instrument for extension professionals to have the structural framework necessary to support and disseminate horticulture technologies and practices. Extension represents a divergent and complex example of a client-serving occupation. Extension provides valuable services and it is imperative the public recognizes extension as a legitimate profession [5]. However, the recognition of extension as a profession requires development of network professionalization capacities

and multifaceted strategies to achieve the desired perception [17]. The items resulting from the proposed extension professionalization scale illustrate how network activities can be integrated into a model of professionalization [19] and offer extension providers practical guidance on professionalization efforts. Additionally, the scale provides a tool for additional scholarship related to extension networks. For both academic and practitioner audiences the extension network professionalization scale may be a beneficial tool to aid in the transformation of the perception of extension as somewhat informal occupation in some locations [26,31,32,34] to a standardized and rigorous profession providing essential, exclusive, and complex services [19]. In parallel with such recognition changes, the services offered to clientele, including those for horticultural producers, should be elevated with the potential to better serve the needs of clientele while helping adopt new advances and technologies.

**Author Contributions:** Conceptualization, K.W.L., A.J.L. and K.D.; methodology, K.W.L., A.J.L. and K.D.; software, K.W.L. and A.J.L.; validation, K.W.L. and A.J.L.; formal analysis, K.W.L. and A.J.L.; investigation, K.W.L. and A.J.L.; resources, K.W.L., A.J.L. and K.D.; data curation, K.W.L. and K.D.; writing—original draft preparation, C.S., K.W.L. and A.J.L.; writing—review and editing, C.S., A.P., K.W.L., A.J.L., K.D. and J.P.; visualization, J.P.; supervision, K.W.L.; project administration, K.W.L., A.J.L., K.D.; funding acquisition, K.W.L., A.J.L. and K.D.; Abstract, J.P. All authors have read and agreed to the published version of the manuscript.

**Funding:** This research was funded by the Global Forum for Rural Advisory Services.

**Data Availability Statement:** Not applicable.

**Acknowledgments:** Not applicable.

**Conflicts of Interest:** The research being reported in this publication was financially supported by the Global Forum for Rural Advisory Services (GFRAS). Two of the authors of this publication served as consultants to GFRAS, and a third author was employed by GFRAS at the time the data were collected. Furthermore, this work was undertaken as part of the CGIAR Research Program on Policies, Institutions, and Markets (PIM), led by the International Food Policy Research Institute (IFPRI). The opinions expressed here belong to the authors, and do not necessarily reflect those of PIM or CGIAR. We have disclosed these interests fully to the journal and have in place a plan for managing any potential conflicts arising from this arrangement.

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
