# Peer review of "Extension Networks and Dissemination of Horticultural Advancements: Development and Validation of a Professionalization Instrument"

_horticulturae, doi:10.3390/horticulturae9020245_

Round 1

Reviewer 1 Report

Regarding this article, I have the following recommendations:

1. The Abstract must have a Goal, Method, Results, and Originality. It is necessary to explain each of them separately.

2. As a general observation, the references in-text are placed at the end of the statement. Including the author(s) as an active subject in the phrase is recommended. 

3. The gap identified in the literature regarding the topic is not clearly explained. We can see in the Introduction section some ideas about the stage in the field. In the article, there is no Literature review as a separate section. It is not present the research questions of the research. The paper's objective is presented in section no. 3, but the research question needs to be more appropriate because it is also a kind of objective. 

4. As a researcher, I'd like to see the objective as the last paragraph in the Introduction. The Conceptual Framework section is redundant for specialists in the field. Do you consider it necessary to explain all the concepts in the area? We think it is more efficient to insist on the key concepts needed for your research if it is opportune. 

5. We recommend in the Methods section include a Model of Research for a good understanding of the research methodology and with impact regarding the replicability of the research. It needs to be clarified whether the number of respondents is or is not significant for the research. Please, insist on this aspect!

6. The statistical explanations are too developed compared to the descriptions of the significance of the results for the field of research. 

7. Including "Discussion" in the "Conclusions, ... " section is not recommended. For this, there is a section before the Conclusions.

Author Response

Thank you for the opportunity to revise and resubmit the manuscript Extension Networks and Dissemination of Horticultural Advancements: Development and Validation of a Professionalization Instrument. The feedback received from the reviewers was constructive and significantly improved the content and quality of the resubmitted manuscript. The specific actions we have taken to address the reviewer’s concerns are detailed below.

  1. The Abstract must have a Goal, Method, Results, and Originality. It is necessary to explain each of them separately.
    1. ACTION: The reviewer feedback is appreciated. Based on recommendations the article abstract has been revised and updated to specifically identify the goal, method, results, and originality of the study.
  2. As a general observation, the references in-text are placed at the end of the statement. Including the author(s) as an active subject in the phrase is recommended. 
    1. ACTION: The reviewer input and feedback is helpful, where possible and appropriate in phrase citations have been added or modified throughout the manuscript.
  3. The gap identified in the literature regarding the topic is not clearly explained. We can see in the Introduction section some ideas about the stage in the field. In the article, there is no Literature review as a separate section. It is not present the research questions of the research. The paper's objective is presented in section no. 3, but the research question needs to be more appropriate because it is also a kind of objective. 
    1. ACTION: We appreciate the reviewer feedback and suggestions. Based on feedback from both reviewers the introduction section and conceptual framework sections were completely rewritten. The conceptual framework section was removed with appropriate sections reintegrated into the introduction. The literature review and need for the study are therefore more closely integrated.
  4. As a researcher, I'd like to see the objective as the last paragraph in the Introduction. The Conceptual Framework section is redundant for specialists in the field. Do you consider it necessary to explain all the concepts in the area? We think it is more efficient to insist on the key concepts needed for your research if it is opportune. 
    1. ACTION: As described previously, the introduction section was completely rewritten to better integrate the need and background of the study, as well as situate the current study to address existing gaps. As suggested, the objective of the study has been moved to the introduction section.
  5. We recommend in the Methods section include a Model of Research for a good understanding of the research methodology and with impact regarding the replicability of the research. It needs to be clarified whether the number of respondents is or is not significant for the research. Please, insist on this aspect!
    1. ACTION: The reviewer feedback is again appreciated. Clarifications regarding the methods used in the study, including both interpretation of number of respondents and EFA analysis, have been added.
  6. The statistical explanations are too developed compared to the descriptions of the significance of the results for the field of research. 
    1. ACTION: The feedback provided is appreciated. Based on the recommendations from both reviewers modifications to the results section have included the removal of results not central to the research. Additionally, descriptions of the significance of the results have been added and clarified in the discussion section.
  7. Including "Discussion" in the "Conclusions, ... " section is not recommended. For this, there is a section before the Conclusions.
    1. ACTION: Based on the provided feedback the discussion and conclusion sections have been separated. Additional content has also been added to the sections to improve the overall utility of the study.

Reviewer 2 Report

While the paper is relevant for the targeted journal there are some issues that need to be addressed before the paper can be considered for publication.

1. Please, in the introduction of your submitted manuscript, describe the difference and value added of this paper compared with your 2021 paper "Perceptions of knowledge management capacity within extension services: An exploratory factor analysis approach".  Where are differences, where does it offer additional insights and which new insights are there?

2. Your conceptual framework (section 2) is quite abstract and theoretical. Please try to integrate practical examples from horticulture into each of your sub-sections of 2.2.

3. Within section 4.1 (data collection) please add information on where you collected your data (which countries), when, and the sampling approach (why theses countries and the networks were selected).

4. The results description (section 5) needs to be improved.  What exactly is the reason for including an "overall extension professionalization index" in your EFA?  Why was there no factor rotation to improve the clarity of the results?

5. In the discussion, please describe in concrete terms how the suggested scale can be used in practice in the field of horticulture.

Author Response

Thank you for the opportunity to revise and resubmit the manuscript Extension Networks and Dissemination of Horticultural Advancements: Development and Validation of a Professionalization Instrument. The feedback received from the reviewers was constructive and significantly improved the content and quality of the resubmitted manuscript. The specific actions we have taken to address the reviewer’s concerns are detailed below.

  1. Please, in the introduction of your submitted manuscript, describe the difference and value added of this paper compared with your 2021 paper "Perceptions of knowledge management capacity within extension services: An exploratory factor analysis approach".  Where are differences, where does it offer additional insights and which new insights are there?
    1. ACTION: Based on the reviewer feedback a clarifying statement has been added to the introduction section. The context of two manuscripts as integrated, but not duplicative, has been clarified.
  2. Your conceptual framework (section 2) is quite abstract and theoretical. Please try to integrate practical examples from horticulture into each of your sub-sections of 2.2.
    1. ACTION: The reviewer feedback is appreciated. Based on both reviews a complete re-write of the introduction and conceptual framework sections was completed. The conceptual framework section was removed with appropriate content integrated into the introduction as suggested. Specifically, numerous practical horticulture examples have been added to the section.
  3. Within section 4.1 (data collection) please add information on where you collected your data (which countries), when, and the sampling approach (why theses countries and the networks were selected).
    1. ACTION: The reviewer feedback for clarification is noted and appreciated. Additional details regarding the convenience nature of the sample and the specific networks in the study has been included. The specific countries (home location of respondents) was not included as this represented 56 countries.
  4. The results description (section 5) needs to be improved.  What exactly is the reason for including an "overall extension professionalization index" in your EFA?  Why was there no factor rotation to improve the clarity of the results?
    1. ACTION: The reviewer feedback is appreciated as it relates to results section of the manuscript. In response to the feedback, the overall index results were removed from the section. This helps to improve clarity and interpretability of the presented results. Additionally, a clarifying statement and citation was added to clarify a varimax rotation was used during the analysis.
  5. In the discussion, please describe in concrete terms how the suggested scale can be used in practice in the field of horticulture.
    1. ACTION: Again, the reviewer feedback is appreciated and insightful. In response to the feedback a new section, ‘Contributions to Horticultural Practice’ was added to the discussion section. The new section provides specific recommendations for how and why extension networks should use this proposed scale, how by doing so networks should improve the rigor of associated extension professionals, and how more professional extension practitioners should be more effective in serving clientele.

Round 2

Reviewer 2 Report

The text has significantly improved.  However, as previously requested, please add the year of data collection to your section 2.1.

The connection and value-added of this study to your previous (2021) paper on the topic could have been made more explicit.  

Nevertheless, overall, the paper can now be considered for publication.

Author Response

  1. The text has significantly improved. However, as previously requested, please add the year of data collection to your section 2.1.
    1. ACTION: The reviewer feedback is appreciated. Apologies for the oversight in addressing the previous feedback, the dates for data collection were added to the methods section of the manuscript.
  2. The connection and value-added of this study to your previous (2021) paper on the topic could have been made more explicit.  
    1. ACTION: Additional narrative has been added to the discussion section to more explicitly connect the results of the current study with the previous research.